# The Antibacterial Effect of Silver Nanoparticles on *Staphylococcus Epidermidis* Strains with Different Biofilm-Forming Ability

**DOI:** 10.3390/nano10051010

**Published:** 2020-05-25

**Authors:** Denis Swolana, Małgorzata Kępa, Danuta Idzik, Arkadiusz Dziedzic, Agata Kabała-Dzik, Tomasz J. Wąsik, Robert D. Wojtyczka

**Affiliations:** 1Department of Microbiology and Virology, Faculty of Pharmaceutical Sciences in Sosnowiec, Medical University of Silesia, ul. Jagiellońska 4, 41-200 Sosnowiec, Poland; deswolana93@wp.pl (D.S.); mkepa@sum.edu.pl (M.K.); didzik@sum.edu.pl (D.I.); twasik@sum.edu.pl (T.J.W.); 2Department of Conservative Dentistry with Endodontics, Faculty of Medical Sciences in Zabrze, Medical University of Silesia, Pl. Akademicki 17, 41-902 Bytom, Poland; adziedzic@sum.edu.pl; 3Department of Pathology, Faculty of Pharmaceutical Sciences in Sosnowiec, Medical University of Silesia, Sosnowiec, ul. Ostrogórska 30, 41-200 Sosnowiec, Poland; adzik@sum.edu.pl

**Keywords:** antimicrobial activity, biofilm, nanosilver, *Staphylococcus epidermidis*

## Abstract

Among many infectious diseases, infections caused by pathogens of *Staphylococcus* species exert a substantial influence upon human health, mainly due to their continuous presence on human skin and mucous membranes. For that reason, an intensive search for new, effective anistaphyloccocal agents can currently be observed worldwide. In recent years, there has been growing interest in nanoparticles, as compounds with potential antibacterial effect. The antibacterial activity of silver containing substances has been well recognized, but thoughtful studies focused on the effect of silver nanoparticles on bacterial biofilm are scarce. The aim of this study was to assess the influence of silver nanoparticles (AgNPs) with particle sizes in the range between 10 and 100 nm, and a concentration range from 1 to 10 µg/mL, upon *Staphylococcus epidermidis* strains with different biofilm-forming abilities (BFAs). The studies revealed the highest level of antimicrobial activity for AgNPs in relation to *S. epidermidis* strains with BFA, and what is more, the observed effect was proportional to the increasing particles’ size, and strains not forming biofilm were more susceptible to silver nanoparticles with the smallest examined size, which was 10 nm.

## 1. Introduction

*Staphylococcus epidermidis* is part of the natural bacterial flora in humans, being the most common species of the human skin microbiome, as well as the respiratory system and alimentary tract. In recent years, it has been considered an important etiological factor, mainly for nosocomial infections and infections related to the cardiovascular system. Those infections affect mainly immunocompromised patients, as well as persons with prosthetic implants made of plastics. The treatment of such infections caused by *S. epidermidis* is difficult due to the increased resistance of those bacteria to numerous groups of antibiotics, and their specific biofilm-forming ability (BFA), which reduces the penetration of antibiotics and thus limits their therapeutic efficacy [1,2]. Coagulase-negative staphylococci (CNS), to which *S. epidermidis* belongs, do not produce so many toxins and enzymes as *Staphylococcus aureus*. Their most significant virulence factor is the ability to form biofilm [2,3]. They deposit polymer substances (exopolysacharide slime, adhesines, collagen-binding protein) on the surface of catheters or other biomaterials, which are the basis for biofilm formation [4,5]. Biofilm is a population of bacteria, growing on a specific surface (biotic, e.g., tissues, or abiotic, e.g., a catheter), which constitutes a mere 10% to 15% of its total volume, and is surrounded by extracellular matrix, composed of sugars, proteins, and extracellular DNA (eDNA), which make up the remaining 85%–90% of the whole biofilm structure. Infections with staphylococci that produce biofilm are difficult to cure, since the bacteria living there may be up to 1000 times more resistant to antibiotic therapy than the pathogens growing in planktonic form [6,7].

The BFA of bacteria hinders the action of chemotherapeutic agents—the only currently available effective therapy against micro-organisms—and results in substantial increase of bacteria resistance. For that reason, the search for the new substances which can affect the cells of microorganisms is ongoing in many laboratories. In recent years, the nanoparticles of metals have been examined as compounds effectively inhibiting micro-organism action [8]. Other than copper and iron oxides, nanosilver has been demonstrated to have the most versatile application in fighting micro-organisms and it has been used for that purpose for quite some time. The antibacterial activity of nanosilver against the strains of *Staphylococcus aureus*, *Pseudomonas aeruginosa,* and *Escherichia coli* has already been demonstrated [8,9,10]. The application of silver as antibacterial agent was possible among other things due to its low cytotoxicity towards mammalian cells, and nanoparticle form increases the possibilities of its application even further [10]. The research conducted by Franci et al. [11] also revealed the significant influence of shape on the activity of silver nanoparticles. Nano-prism molecules demonstrated higher activity than spherical nanoparticles [12]. The additional advantage of nanosilver lies in its wide spectrum of bactericidal or bacteriostatic activity at low concentrations, without triggering resistance mechanisms in bacteria [13].

To date, to the best of our knowledge, there have been no reports on the influence of silver nanoparticles (AgNPs) on *S. epidermidis,* bacteria, responsible for the significant number of biofilm-related infections associated with human implant surgical procedures. The rapidly growing number of biofilm-forming *S. epidermidis* infections, as well as the antibiotic-resistance development of the pathogenic bacterial strains, generates the need for thorough recognition of the mechanism(s) of nanosilver action on the *S. epidermidis* strains, as well as the evaluation of possible applications of AgNPs as anistaphylococcal agent.

## 2. Results and Discussion

### 2.1. The Assessment of AgNPs’ Anti-Staphyloccocus Activity

The antibacterial properties of AgNPs have been known for years, but the discussion is still going on regarding the mechanism responsible for their bactericidal effect. It has not been fully proven whether the AgNPs or silver ions originating in the cell environment cause the cytotoxic effect on bacterial cells. Undoubtedly, it is related to the direct contact of AgNPs with the bacterial cell wall, and the penetration to cell cytoplasm [14]. Wang et al. [15] proved that the highest cytotoxic activity was demonstrated by AgNPs with sizes below 10 nm. In this study, we assessed the activity of AgNPs in the size range from 10 nm to 100 nm on three standard strains of *S. epidermidis.* Our data demonstrated that the activity of AgNPs towards *S. epidermidis* strains (both those generating biofilm and those not generating it) was noted only in case of small AgNPs, with the size ranged between 10 and 40 nm (Table 1).

Other authors showed that bactericidal effect is also caused by the contact of AgNPs with the enzymatic system of bacterial cells and reactive oxygen species (ROS) generation. The consequence of increased ROS activity is oxidative stress, causing damage to proteins and nucleic acids [14]. The study conducted by Sondi and Salopek-Sondi [16] indicated that the accumulation of AgNPs of 12 nm size in the cell wall of *Escherichia coli* led to the formation of indentations/cavities, which in turn caused the loss of external cell membrane integrity and resulted in cell death. Similar studies on the size of AgNPs and their influence on *E. coli* strains were conducted by Morones et al. [17]. They showed that AgNPs with particle sizes of 1–10 nm were attached/anchored to the cell wall of *E. coli* and interfered with its normal functions, such as permeability and respiration. Gahlawat et al. [18], in turn, demonstrated that AgNPs with the size of 10 nm were attached to the wall of *V. cholerae* cells, thus disturbing cell permeability and metabolic pathways, causing cell apoptosis. Different bacterial strains showed different susceptibility to AgNPs and the effect depends on the type of pathogen. Gram-negative bacteria proved to be more susceptible to AgNPs, mainly because cellular wall structure, which in the case of Gram-positive bacteria has multiple layers of mucopeptides, constituting a barrier protecting particles from penetration into the cytoplasm [14]. However, in our study we demonstrated that the activity of AgNPs with sizes ranging from 10 nm to 100 nm also affected Gram-positive bacteria, such as *S. epidermidis*. This activity differed, depending on the biofilm-forming ability of those micro-organisms. Only the *S. epidermidis* ATCC 35984 strain, with substantial BFA, was sensitive to AgNPs of all the sizes investigated, which may be due to their strong accumulation in the biofilm structure. In the case of all biofilm-forming strains, the most intense activity was demonstrated by nanoparticles with the size from 20 nm to 100 nm. In the case of nanoparticles with a size of 10 nm, their intense activity in relation to *S. epidermidis* ATCC 12228, not forming biofilm, was also demonstrated. Nanda et al. [19] also conducted interesting studies, as they examined the influence of silver bio-nanoparticles synthesized with the use of wild-type *S. aureus* upon various pathogenic bacteria, among other the MRSA and MRSE strains. They demonstrated a more intense activity of the bio-nanoparticles obtained in relation to Gram-positive bacteria than was the case for Gram-negative bacterial strains. The bio-nanoparticles thus obtained are cheap and not harmful/toxic for the environment. This method makes it possible to generate large-diameter nanoparticles of 160 and 180 nm. In the studies conducted by Lu et al. [20], the activity of AgNPs of size 5 nm was also demonstrated, in case of pretty diverse bacterial flora of the oral cavity. The studies demonstrated that the bactericidal influence of AgNPs depended not only on particle size, but also on shape, surface charge, dose, and dispersion of the particles. Well dispersed AgNPs in physiologic salt solutions demonstrated higher bactericidal efficiency than agglomerates [14]. Archaya et al. [21] examined the bactericidal activity of nanosilver with a spherical or oblong shape on Gram-positive (*S. aureus*, *B. subtilis*) and Gram-negative (*E. coli*, *K. pneumoniae*, *P. aeruginosa*) strains. Their studies demonstrated that the bactericidal activity in the case of both shapes of AgNPs depended on the dose and time.

It has been demonstrated that another important parameter in the assessment of AgNPs activity is the particle concentration [22,23,24,25]. In the present studies on the AgNPs with sizes between 10 nm and 100 nm, anti-staphylococcal activity was observed for the concentrations ranging from 1 to 10 µg/mL, with the percentage reduction of micro-organisms being determined in reference to *S. epidermidis* strains not exposed to nanosilver (Figure 1). The highest percentage of reduction for micro-organisms was observed in the case of AgNPs having the size of 10 nm; in the case of the *S. epidermidis* ATCC 12228 strain not forming biofilm, the reduction reached as high as 76%. In the case of bigger AgNPs, the reduction percentage with respect to micro-organisms was less profound (up to 58%) and was more pronounced for strains that formed biofilm.

The assessment of concentration influence upon the anti-bacterial activity of AgNPs was also studied by Yu-Guo et al. [22]. The authors examined the influence of AgNPs on *S. aureus* and *P. aeruginosa* strains. Their results confirmed the dose-dependent effect exerted by AgNPs on bacteria. In the case of *S. aureus* strain, the MIC value was 2 µg/mL, whereas the MBC (minimal bactericidal concentration) was twice as high, and amounted to 4 µg/mL. For the *Pseudomonas aeruginosa* strain, both values were lower by half, which supports the notion of greater potency of AgNPs towards cells of Gram-negative bacteria. The average size of AgNPs in this experiment was 11 nm, and thus these results are comparable with those reported in our study [22]. In the study by Gurunathan et al. [23], it was found that in case of the *S. aureus* strain, the lowest concentration inhibiting the growth of micro-organisms in case of AgNPs with diameter of 5 nm, amounted to 0.75 µg/mL. The assessment of cell survivability/viability reported by that research team for the concentration ranged from 0 to 10 µg/mL, revealing a decrease by the factor of about three in the number of bacteria in that range, which is the same as the results we obtained. In that range of concentrations, the number of units forming colonies also dropped about three times. Additionally, the comparison presented by Gurunathan et al. of results obtained for Gram-positive and Gram-negative bacteria revealed the activity to be lower by half for the same concentration of nanosilver in the case of Gram-positive bacteria. The results obtained also suggest that AgNPs with smaller particle size may provide better bactericidal effect, and can also act on an area whose size is bigger, in comparison with AgNPs of bigger diameter [23].

Different results have been shown by Sheikholeslami et al. [24], who assessed the value of the minimal inhibitory concentration (MIC) for *S. epidermidis* strain to be at the level of 62.5 µg/mL in case of AgNPs with a size of about 40 nm. As can be noted, the activity was demonstrated, but at a very high MIC value. In the study conducted by Habash et al. [25], the influence of AgNPs with sizes from 10 nm to 100 nm upon Gram-negative bacteria was assessed. The researchers confirmed a more pronounced activity of nanosilver against Gram-negative bacteria (and their biofilms), with MIC values even lower than those obtained in our study. In summary, it should be noted that AgNPs have demonstrated more pronounced activity against the *S. epidermidis* ATCC 35984 strain, possessing BFA, which may be associated with the accumulation of those particles in biofilm structures. In case of non-biofilm-forming strains, small AgNPs were the most active ones.

### 2.2. Determination of MBIC for AgNPs

In a further stage of the studies, minimal biofilm inhibitory concentration (MBIC) was determined for *S. epidermidis* strains, depending on the nanosilver particle size. For the *S. epidermidis* ATCC 35983 strain (with average BFA), MBIC value for AgNPs with sizes of 10 nm amounted to 9 µg/mL. In the case of other sizes of AgNPs (20–100 nm), no inhibition in biofilm formation was detected, on the contrary, enhanced BFA was noted. The *S. epidermidis* ATCC 35984 strain (with superior BFA) demonstrated similarly high level of BFA in case of all concentration ranges and all sizes of AgNPs. The *S. epidermidis* ATCC 12228 strain failed to demonstrate BFA for all particle sizes (Figure 2). In the study conducted by Gurunathan et al. [23], inhibition of biofilm activity was demonstrated for lower concentrations of AgNPs than those which reduced cell viability.

### 2.3. Determination of Bacterial Cell Viability by Alamar Blue Test

The viability of *S. epidermidis* was also determined, under exposure to AgNPs, with sizes of 10 nm and at concentrations of 1–7 µg/mL, over a time from 12 to 24 h, expressed as the reduction rate of resazurin, for three standard strains: *S. epidermidis* ATCC 12228, *S. epidermidis* ATCC 35983, and *S. epidermidis* ATCC 35984 (Figure 3, Figure 4 and Figure 5).

Viability assessment with the use of the Alamar Blue test demonstrated the inhibiting action of AgNPs, for all examined strains. For biofilm-forming strains, the bacteriostatic activity of AgNPs at a concentration of 2 µg/mL decreased even after the 14th hour of examination. The most pronounced activity affecting biofilm-forming strains was observed for the concentration of 7 µg/mL, where after 22–24 h of investigations, no viability of the examined biofilm-forming strains was detected (Figure 3, Figure 4 and Figure 5). The assessment of the viability of strains expose to AgNPs was also performed by Yu-Guo et al. [22]. In their studies, they analyzed the viability of *S. aureus* cells after the application of silver nanoparticles with a particle size of about 13 nm. The results even demonstrated a reduction in viability in the case of *S. aureus* cells, from the 4th to 24th hour after application of AgNPs.

### 2.4. Evaluation of AgNPs Cytotoxicity Toward Eucariotic Fibroblast Cells of the GM03348 Line

To evaluate the cytotoxicity of AgNPs towards eukaryotic cells, we assessed the effect of AgNPs with sizes of 10 nm and 20 nm (active towards all the strains examined) at a concentration of 5 µg/mL (MIC value) on fibroblast cells of the GM03348 line. In the cell culture, the cells were spindle-shaped and characterized by short cytoplasmatic processes (Figure 6A). Agranular cytoplasm was uniformly stained in pink. The cell nucleus was located centrally, with a regular contour and noticeable nucleoli. Cells of GM03348 line adhered strongly to the surface and formed clusters. Additionally, an interesting observation concerned the strong cell–cell junctions. In the case of cells exposed to silver nanoparticles with sizes of 10 nm and 20 nm at a concentration of 5 µg/mL (Figure 6B,C), no changes in morphology were noted under the light microscope. The phenotype of the cells did not reveal necrotic features. Cells of the GM03348 line employed a regular contour, they did not increase their size, and organelles did not become oedematous. In the cell culture, neither cytoplasmatic inclusion nor cytoplasmic vacuolization were revealed. Simultaneously, no nucleus decomposition or fragmentation was observed.

### 2.5. Statistical Analysis

Statistical analysis of absorbance measurements after exposing the *S. epidermidis* ATCC 12228 strain to AgNPs, regardless of the concentration, revealed the existence of a statistically significant dependence on the size of the AgNPs (*p* < 0.05). At the same time, regardless the size of the silver AgNPs, a statistically significant relationship with concentration was demonstrated (*p* < 0.05). Additionally, the interaction between the concentration and the size of AgNPs reached statistical significance (*p* < 0.05). In terms of post hoc analysis, the Dunnett test was performed for each size of AgNPs. Zero concentration of AgNPs was considered as a control. The results of the analysis for AgNPs with a size of 10 nm demonstrated the existence of a statistically significant relationship in the concentration range of 5 µg/mL to 10 µg/mL in comparison with the control. In the case of a particle size of 40 nm, statistical significance was demonstrated for the concentration range from 6 µg/mL to 10 µg/mL in comparison with control, whereas for silver AgNPs with a size of 100 nm, statistical significance was observed for the concentration range of 8–10 µg/mL. For *S. epidermidis* ATCC 35983, statistical analysis revealed the existence of statistically significant dependence on AgNPs size (*p* < 0.05), irrespective of concentration. At the same time, statistically significant dependence on concentration was demonstrated (*p <* 0.05), irrespective of the size of the AgNPs. However, no statistically significant interaction was found to exist between the concentration and the size of the AgNPs (*p* > 0.05). In terms of post hoc analysis, the Dunnett test was performed for each size of the silver nanoparticles. Zero concentration of nanosilver was assumed as control. The results of the analysis for AgNPs having the size of 10 nm revealed the existence of statistically significant dependence for the concentration of 10 µg/mL in comparison with the control. In the case of a particle size of 40 nm, statistical significance was demonstrated for the concentration range of 8 µg/mL to 10 µg/mL in comparison with the control, whereas for AgNPs with a size of 100 nm, statistical significance occurred in the range of 4 µg/mL to 10 µg/mL.

The statistical analysis performed for *S. epidermidis* ATCC 35984 revealed the existence of statistically significant dependence on nanoparticle size (*p* < 0.05), irrespective of concentration. At the same time, statistically significant dependence on concentration was demonstrated (*p* < 0.05), irrespective of the size of AgNPs. Additionally, interaction between the concentration and the size of the AgNPs was demonstrated (*p* > 0.05). In terms of post hoc analysis, the Dunnett test was performed for each size of AgNPs. Zero concentration of nanosilver was assumed as the control. The results of analysis for AgNPs with a of 10 nm revealed the existence of statistically significant dependence in the concentration range of 5 µg/mL to 10 µg/mL in comparison with control. In the case of particles with a size of 40 nm, statistical significance was demonstrated for the concentration range from 6 µg/mL to 10 µg/mL in comparison with control, while for AgNPs with a size of 100 nm, statistical significance occurred in the range from 2 µg/mL to 10 µg/mL.

## 3. Experimental Section

### 3.1. Bacterial Strains, Media, and Reagents

The second passage of three reference strains was used for investigations, namely: *S. epidermidis* ATCC 12228, a strain that does not have phenotype biofilm-forming ability, *S. epidermidis* ATCC 35983, a strain possessing moderate biofilm-forming ability and having genes of operon *icaADBC*, and *S. epidermidis* ATCC 35984, a strain with substantial biofilm-generating ability and possessing genes of operon *icaADBC.* Bacterial strains were stored in Tryptic Soy Broth (TSB) medium with the addition of 20% glycerol, in −86 °C. Silver nanoparticles with sizes of 10 nm, 20 nm, 40 nm, 60 nm and 100 nm (TEM), 0.02 mg/mL in aqueous buffer, containing sodium citrate as stabilizer (Sigma-Aldrich/Merck—Darmstadt, Germany), were used in the study.

### 3.2. Determination of Antistaphylococcal Activity of AgNPs

The antimicrobial activity of AgNPs was determined using the broth microdilution method (BMM), as recommended by the National Committee for Clinical Laboratory Standards. Minimal inhibitory concentration (MIC) of AgNPs having the following sizes: 10, 20, 40, 60, 100 nm, was measured for three reference strains of *S. epidermidis* in TSB medium supplemented with 0.25% of glucose. The examined range of AgNPs concentrations was between 0.5 µg/mL and 20 µg/mL. The microtitration plates prepared underwent incubation at a temperature of 37 °C for 24 h; subsequently, the absorbance was determined by means of spectrophotometry, wavelength *λ* = 595 nm (Multiscan EX, Thermo Electron Corporation, Shanghai, China), in order to determine the MIC value. Each plate contained growth control as well as sterility control. The investigations were repeated six times. The percentage of growth reduction is calculated using the formula in Equation (1):(1)$$GR\;\left[\%\right]=\frac{OD_{S}-OD_{T}}{OD_{S}}\times 100\%.$$
*GR*—percentage of growth reduction*OD_S_*—the average value of spectrophotometric measurements at *λ* = 595 nm for the control sample without the addition of AgNPs*OD_T_*—the average value of spectrophotometric measurements at *λ* = 595 nm for the studied sample with the addition of AgNPs at the studied concentration

### 3.3. Determination of MBIC Values for AgNPs

The examinations of minimal biofilm inhibitory concentration (MBIC) were carried out in accordance with the modified Christensen method [26]. For that purpose, after 24 h of incubation with silver nanoparticles, the wells were washed three times with phosphate buffer (PBS, pH = 7.2) and all the planktonic forms of bacteria were removed. The plates were dried at a temperature of 37 °C, after which 150 µL of 1% crystal violet (MERC/Sigma-Aldrich—Darmstadt, Germany) was added to each well in order to stain the biofilm. The plates were incubated for 30 min at room temperature. The stain was then removed, plates were flushed four times with deionized water and dried at a temperature of 37 °C. The biofilm that was formed was dissolved in 200 μL of 95% isopropanol in 1 M of HCl. After 5-min incubation, samples of 100 μL of the solution were collected, transferred to a new titration plate, and absorbance was measured for the wavelength of *λ* = 490 nm, by means of a Multscan EX (Thermo Electron Corporation, Shanghai, China) microplate reader. The assessment of BFA was performed on the basis of comparing the absorbance values with the *S. epidermidis* ATCC 12228 strain (negative control). The examinations were performed three times.

### 3.4. Determination of Viability of Bacterial Cells by Alamar Blue Test

The assessment of the viability of cells subjected to AgNPs was performed with the use of a test named Alamar Blue Cell Viability Reagent (INVITROGEN – Eugene, Oregon, USA), in accordance with the protocol provided by the producer. Ten µL of Alamar Blue reagent was added to the 96-well titration plate, containing 100 µL of bacteria suspension, exposed to AgNPs of various sizes, and in different concentrations. The entire material was incubated at a temperature of 37 °C for 1 h. Subsequently, absorbance was measured at wavelengths of *λ* = 570 nm and *λ* = 600 nm, by means of a Multiscan EX (Thermo Electron Corporation, Shanghai, China) microplate reader. The examinations were performed three times.

### 3.5. Evaluation of AgNPs’ Cytotoxicity Toward Eucariotic Fibroblast Cells of the GM03348 Line

Hematoxylin-eosin (H-E) stain is a so-called topographic view stain, which makes it possible to assess the entire structure of a cell, thanks to contrast staining of cytoplasm and cell nuclei. Staining was performed with the use of reagents by Avantor Performance Materials (Gliwice, Poland). The cells to be stained were seeded after trypsinization to four-chamber slides, Nunc™ Lab-Tek™ II Chamber Slide™ System (Nunc – Roskilde, Denmark). 1 mL of pure medium was introduced into each chamber, and the slides were put in an incubator. After 24 h of incubation time, the cells adhered to slide surface, and the confluence was assessed to be 80%. The culture medium was gently removed from above the cells, and 1 mL of silver nanoparticles solution, with particle sizes of 10 nm or 20 nm and a concentration 5 µg/mL, was introduced into the chambers. 1 mL of pure medium was pipetted into one chamber (control). The culture was continued for a further 24 h under standard conditions. After the completion of incubation, the medium and chambers were removed, while the examined cells were fixed on the culture dish surface through 12 h incubation in 96% ethanol. Staining was performed in accordance with the producer’s procedure. The morphological image of the cells was obtained using a Zeiss Axiostar microscope (Carl Zeiss, Jena, Germany).

### 3.6. Statistical Analysis

Statistical analysis was performed with the use of Statistica 13.5 PL software (StatSoft, Kraków, Poland). The Shapiro-Wilk test was used to assess the distribution normality, while the Levene test was applied to confirm the variance homogeneity. Subsequently, two-way analysis of variance was performed. For the comparison of results with control, the Dunnett test was also employed.

## 4. Conclusions

Nanosilver demonstrates varying activity in inhibiting the growth and production of biofilm in the reference strains of *S. epidermidis,* and this is related to the size of nanoparticles applied. The highest efficacy towards standard *S. epidermidis* strains was demonstrated by nanosilver with a particle size of 10 nm and a concentration of 5 µg/mL. For strains which form biofilm, the nanosilver activity increased with increasing particle size.

## Figures and Tables

**Figure 1 nanomaterials-10-01010-f001:**
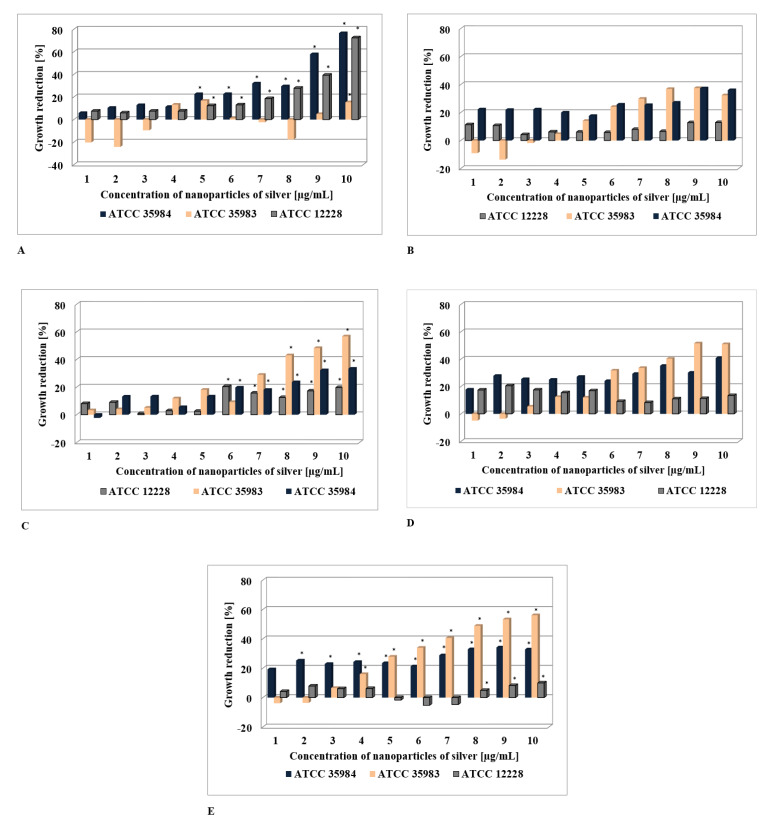
Growth reduction in case of *S. epidermidis* strains subjected to AgNPs, particle size 10 nm (**A**), 20 nm (**B**), 40 nm (**C**), 60 nm (**D**), 100 nm (**E**). The histograms provide the *p*-values obtained by ANOVA analysis. Post-hoc comparison results (Dunnett test, *p <* 0.05), are summarized with asterisks to underline the most relevant differences in Ag-NP-treated samples with respect to the control.

**Figure 2 nanomaterials-10-01010-f002:**
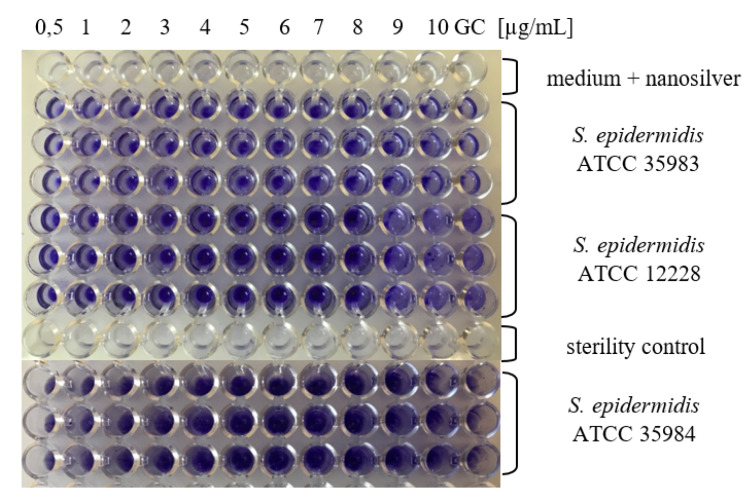
Assessment of BFA of *S. epidermidis* strains exposed to various concentrations of AgNPs, particle size 10 nm. From top to bottom, in rows: row 1: Tryptic Soy Broth (TSB) medium + nanosilver; rows 2–4: TSB medium + nanosilver + standard strain *S. epidermidis* ATCC 12228, rows 5–7: TSB medium + nanosilver + standard strain *S. epidermidis* ATCC 35983; row 8: TSB medium (sterility control); rows 9–11: TSB medium + nanosilver + standard strain *S. epidermidis* ATCC 35984. GC- growth control.

**Figure 3 nanomaterials-10-01010-f003:**
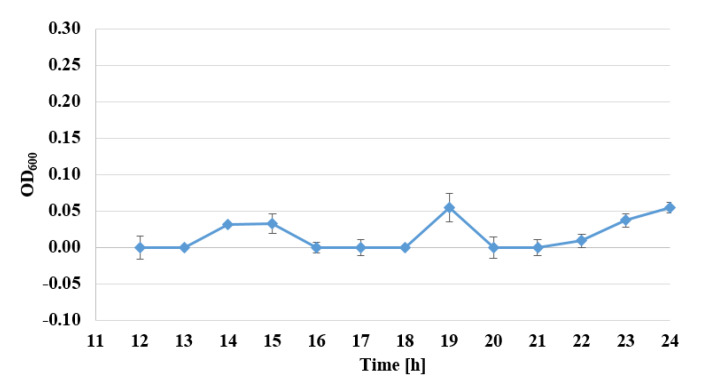
Changes of absorbance in Alamar Blue test for the strain *S. epidermidis* ATCC 35983, exposed to the activity of nanosilver particles with sizes of 10 nm and a concentration of 7 µg/mL between the 12th and 24th hour (*λ* = 600 nm).

**Figure 4 nanomaterials-10-01010-f004:**
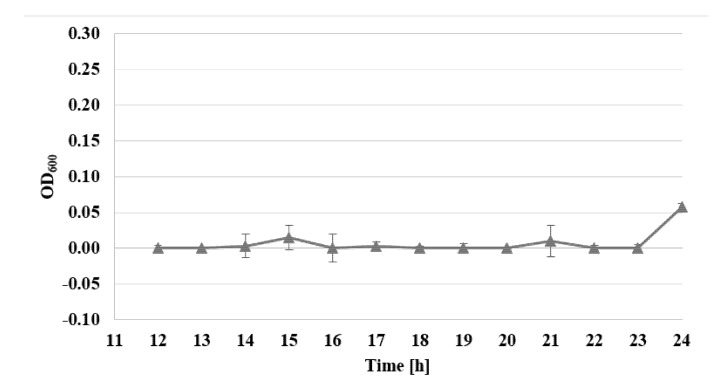
Changes of absorbance in Alamar Blue test for the strain *S. epidermidis* ATCC 35984, exposed to the activity of nanosilver particles with sizes of 10 nm and a concentration of 7 µg/mL between the 12th and 24th hour (*λ* = 600 nm).

**Figure 5 nanomaterials-10-01010-f005:**
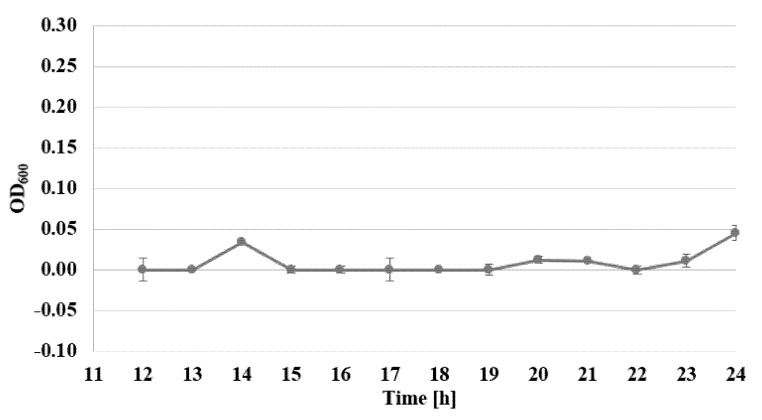
Changes of absorbance in Alamar Blue test for the strain *S. epidermidis* ATCC 12228, exposed to the activity of nanosilver particles with sizes of 10 nm and a concentration of 7 µg/mL between the 12th and 24th hour (*λ* = 600 nm).

**Figure 6 nanomaterials-10-01010-f006:**
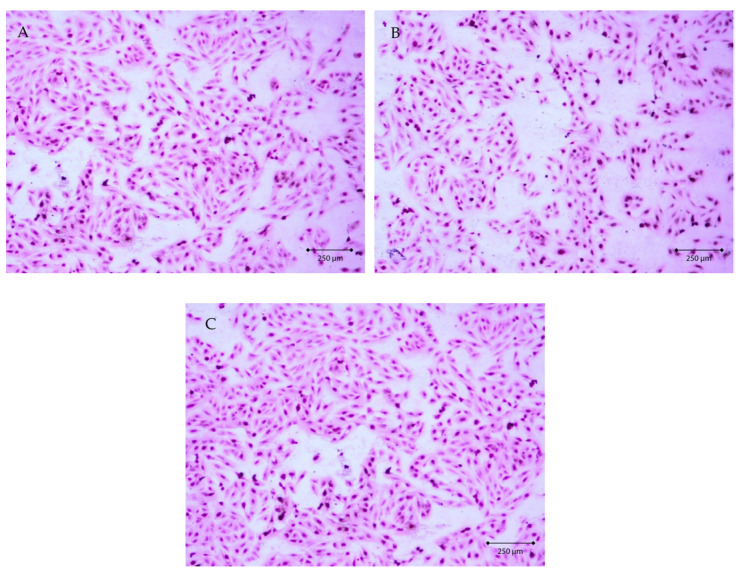
The image of cells of the GM03348 line in hematoxylin-eosin (H-E) stain at 100× magnification: (**A**) control, (**B**) exposed to silver nanoparticles, particle size 10 nm, concentration 5 µg/mL, (**C**) exposed to AgNPs, particle size 20 nm, concentration 5 µg/mL.

**Table 1 nanomaterials-10-01010-t001:** MIC values [µg/mL] for the *S. epidermidis* strains examined, depending on AgNPs particle size; BA—activity not observed.

Nanosilver Particle Size	*S. epidermidis* ATCC 12228	*S. epidermidis* ATCC 36983	*S. epidermidis* ATCC 35984
10 nm	3 µg/mL	4 µg/mL	5 µg/mL
20 nm	BA	4 µg/mL	1 µg/mL
40 nm	6 µg/mL	4 µg/mL	1 µg/mL
60 nm	BA	4 µg/mL	1 µg/mL
100 nm	BA	3 µg/mL	1 µg/mL

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
