# Peer review of "The Antibacterial Effect of Silver Nanoparticles on Staphylococcus Epidermidis Strains with Different Biofilm-Forming Ability"

_nanomaterials, 2020, doi:10.3390/nano10051010_

Round 1

Reviewer 1 Report

The manuscript is well written and well presented. The results presented are meaningful and worth of publication. However, before the work can be accepted for publication, the authors should provide some additional information in the Experimental Section.

  1. Were these commercially sourced silver particles? If so, where were they purchased from? 
  2.  If they are not commercially obtained, how were they synthesized?
  3.  Can the authors comment on the size distribution of the silver particles?
  4.  Can the authors provide a microscopy image of the silver particles?

Author Response

We would like to express our gratitude for detailed and critical reading of the manuscript. We gratefully acknowledge the Reviewer for constructive comments and helpful suggestions. The authors are convinced that introduction of the suggested changes and elucidations in the corrected version of the manuscript will enhance the quality of the presented work. We trust that the Reviewer will find our corrections satisfactory and the manuscript suitable for publication.

Comment 1 and 2.

  1. Were these commercially sourced silver particles? If so, where were they purchased from? 

  2.  If they are not commercially obtained, how were they synthesized?

Authors’ reply: commercially available silver nanoparticles with the size of 10, 20, 40, 60 and 100 nm (TEM assessment, according to producer specification) and 0.02 mg/mL in aqueous buffer, containing sodium citrate as stabilizer, were purchased from Sigma-Aldrich/Merck (Germany) and were used in all experiments. This information has been added to the manuscript  in Section 3.1.

 3.1. Bacterial Strains, Media, and Reagents

The second passage of three reference strains was used for investigations, namely: S. epidermidis ATCC 12228 – a strain that does not have phenotype biofilm-forming ability, S. epidermidis ATCC 35983 – a strain possessing moderate biofilm-forming ability and having genes of operon icaADBC, and S. epidermidis ATCC 35984 – a strain having substantial biofilm-generating ability and possessing genes of operon icaADBC. Bacterial strains were stored in Tryptic Soy Broth (TSB) medium with the addition of 20% glycerol, in -86°C. Silver nanoparticles of the size of 10nm, 20nm, 40nm, 60nm and 100 nm (TEM) 0.02 mg/mL in aqueous buffer, containing sodium citrate as stabilizer (Sigma-Aldrich/Merck - Germany) were used in the study.

Comment 3 and 4

  1.  Can the authors comment on the size distribution of the silver particles?

  2.  Can the authors provide a microscopy image of the silver particles?

Authors’ reply: according to the obtained results, AgNPs of the smallest size used (10 nm) affect both biofilm-forming (ATCC 35984) and planktonic forms of S. epidermidis reference strains. However nanoparticles of diameter of 20 nm and more, AgNPs demonstrate more potent activity for those strains possibly due to binding into the biofilm structure, what was reflected in decreased MIC values. The exact mechanism is currently under investigations, with the use of scanning electron microscopy methods (SEM), among others. These results will be presented in the subsequent publication in the cycle and all relevant  images will be published in a forthcoming paper.

Reviewer 2 Report

Authors have done intensive work on different size and concentration of silver nanoparticles on Biofilm of Staphylococcus epidermidis.

Introduction

Nanda has found that the most sensitive antimicrobial activity of silver nano particles has been observed against methicillin-resistant S. aureus followed by methicillin-resistant Staphylococcus epidermidis.

Anima Nanda, M.Saravanan.  Biosynthesis of silver nanoparticles from Staphylococcus aureus and its antimicrobial activity against MRSA and MRSE. Nanomedicine: Nanotechnology, Biology and Medicine Volume 5, Issue 4, December 2009, Pages 452-456 https://doi.org/10.1016/j.nano.2009.01.012

You can compare the sizes of your silver nano particle and toxicity effect with the existed work.

Method

It seems results of figure 1 came out from method 3,2 and 3,4 . However, there is not connection between the measurement to figures. Please describe how to convert the density of spectrophotometer of 3.2 and 3.4 to percentage of growth.

Result

Line 214 what is the unite of concentration of 5 μm/ml?

Figure 1

please use same font size in the figure 

Figure 3-5

Can you quantify density of spectrophotometer of your figure and convert data to the graph which will be easy to see the time cost?

Figure 6 the scale bars are missing.

Author Response

We would like to express our gratitude for detailed and critical reading of the manuscript. We gratefully acknowledge the Reviewer for constructive comments and helpful suggestions. The authors are convinced that introduction of the suggested changes and elucidations in the corrected version of the manuscript will enhance the quality of the presented work. We trust that the Reviewer will find our corrections satisfactory and the manuscript suitable for publication.

Comment 1

  1. Introduction

Nanda has found that the most sensitive antimicrobial activity of silver nano particles has been observed against methicillin-resistant S. aureus followed by methicillin-resistant Staphylococcus epidermidis. Anima Nanda, M. Saravanan.  Biosynthesis of silver nanoparticles from Staphylococcus aureus and its antimicrobial activity against MRSA and MRSE. Nanomedicine: Nanotechnology, Biology and Medicine Volume 5, Issue 4, December 2009, Pages 452-456 https://doi.org/10.1016/j.nano.2009.01.012

You can compare the sizes of your silver nano particle and toxicity effect with the existed work.

Authors’ reply: as recommended, this reference proposed by the reviewer was added to the manuscript content [19] and discussed accordingly, regarding in vitro toxicity effect. We do appreciate a valuable suggestion made by Reviewer, and we are convinced that it enhances results comparison:

„Also Nanda et al. [19] conducted interesting studies, as they examined the influence of silver bio-nanoparticles synthesized with the use of wild-type S. aureus, upon various pathogenic bacteria, among other the MRSA and MRSE strains. They have demonstrated a more intense activity of the bio-nanoparticles obtained in relation to Gram-positive bacteria, than was the case for Gram-negative bacterial strains. Bio-nanoparticles thus obtained are cheap and not harmful/toxic for the environment. This method allows to generate large diameter nanoparticles of 160 and 180 nm “

Comment 2

  1. Method

It seems results of figure 1 came out from method 3,2 and 3,4 . However, there is not connection between the measurement to figures. Please describe how to convert the density of spectrophotometer of 3.2 and 3.4 to percentage of growth.

Authors’ reply: in Methods section, the principles of conversion from the density of spectrophotometer to percentage of strains growth was added as requested (Section 3.2)

Comment 3

  1. Result
  • Line 214 what is the unit of concentration of 5 μm/ml?

Authors’ reply: the unit’s error has been corrected to 5 μg/ml. Thank you for notifying us about this incorrect value:

“2.4. Evaluation of AgNPs cytotoxicity toward eukaryotic fibroblast cells of the GM03348 line

To evaluate cytotoxicity of AgNPs towards eukaryotic cells we assessed the effect of AgNPs of size of 10 nm and 20 nm (active towards all the strains examined) in the concentration of 5 µg/ml (MIC value) on fibroblast cells of the GM03348 line. In the cell culture, the cells were spindle-shaped and characterized by short cytoplasmatic processes (Figure. 6A). Agranular cytoplasm was uniformly stained in pink. The cell nucleus was located centrally, with a regular contour and noticeable nucleoli.”

  • Figure 1 please use same font size in the figure 

Author’s reply: the font size in the figure 1C has been standardized as suggested.

  • Figure 3-5

Can you quantify density of spectrophotometer of your figure and convert data to the graph which will be easy to see the time cost?

Authors’ reply: density value was implemented as suggested, and Figures 3-5 were generated following suggested conversion to line graphs.

  • Figure 6 the scale bars are missing.

Authors’ reply: Figure 6 has been supplemented with a scale bars as requested.

Round 2

Reviewer 2 Report

The new manuscript has been improved. The section of 2.5. Statistical analysis is difficult to track with long test. Can you please indicated the statistical results in the graph figure 1 and figure 4? please indicate which groups are comparing and p<0.05 with a star '*' in each graph.

Author Response

We would like to express our gratitude for detailed and critical reading of the manuscript. We gratefully acknowledge the Reviewer for constructive comments and helpful suggestions.

Reviver coment

The new manuscript has been improved. The section of 2.5. Statistical analysis is difficult to track with long test. Can you please indicated the statistical results in the graph figure 1 and figure 4? please indicate which groups are comparing and p<0.05 with a star '*' in each graph.

Authors’ reply:

We've added the results of the statistical analysis to Figure 1 and marked it with an asterisk. We supplemented the description of Figure 1 by adding a sentence “The histograms provide the P-values obtained by ANOVA analysis. Post-hoc comparison results (Dunnett test, p<0.05), are summarized with asterisks to underline the most relevant differences in Ag-NP-treated samples with respect to the control.”

Statistical analysis of Figs 3-5 was not conducted and is not described in section 2.5. The previous versions were presented as photographs.